# Seasonal and Interannual Variability of Sea Surface Salinity Near Major River Mouths of the World Ocean Inferred from Gridded Satellite and In-Situ Salinity Products

**Severine Fournier *** and **Tong Lee**

Jet Propulsion Laboratory, California Institute of Technology, Pasadena, CA 91109, USA; tong.lee@jpl.nasa.gov
* Correspondence: severine.fournier@jpl.nasa.gov

**Abstract:** Large rivers are key components of the land-ocean branch of the global water and biogeochemical cycles. River discharges can have important influences on physical, biological, optical, and chemical processes in coastal oceans. It is, therefore, of importance to routinely monitor the time-varying dispersal patterns of river plumes. The European Space Agency (ESA) Soil Moisture and Ocean Salinity (SMOS) and the NASA Soil Moisture Active Passive (SMAP) satellites provide Sea Surface Salinity (SSS) observations capable of characterizing the spatial and temporal variability of major river plumes. The main objective of this study is to examine the consistency of SSS products, from these two missions, and two in-situ gridded salinity products in depicting SSS variations on seasonal to interannual time scales within a few hundred kilometers of major river mouths. We show that SSS from SMOS and SMAP satellites have good consistency in depicting seasonal and interannual SSS variations near major river mouths. The two gridded in-situ products underestimate these variations substantially. This underestimation, most notably associated with the low SSS season following the high-discharge season, is attributable to the limited in-situ sampling of the river plumes when they are the most active. This work underscores the importance of using satellite SSS to study river plumes, as well as to evaluate and constrain models.

**Keywords:** sea surface salinity; river plumes; SMAP; SMOS; Argo

## 1. Introduction

Large rivers are key components of the land-ocean branch of the global water and biogeochemical cycles. Although river runoff represents only 10% of the total freshwater input to the ocean [1], it can have important influences on physical, biological, optical, and chemical processes in the coastal oceans. Large riverine freshwater inputs into the ocean affects ocean currents and air–sea interactions through their effects on near-surface stratification, including the formation of barrier layer that isolates colder subsurface waters from warmer surface waters. The effect on stratification and mixing also has implications for ecosystems and biogeochemistry, for instance, by regulating the nutrient supply from the subsurface to surface waters and affecting the biological activity in the surface layer [2–4]. Rivers also provide organic and inorganic matter, nutrients, sediments into the ocean, impacting the biological and ecological activity in the coastal ocean [5,6]. It is therefore of importance to routinely monitor the dispersal patterns of river plumes at time scales from weekly, monthly, seasonally, to interannually.

Sea Surface Salinity (SSS) observations have been extensively used to trace major river plumes into the ocean [7–14]. Satellite measurements of SSS have been available from the Soil Moisture and Ocean Salinity (SMOS) satellite of the European Space Agency (ESA) since 2010 and from the Soil Moisture Active Passive (SMAP) satellite of the National Aeronautics and Space Administration (NASA) since 2015. These observations allow us to get global measurements of SSS at enhanced spatial and temporal resolutions. SMOS

SSS measurements have a ~50-km spatial resolution, an 18-day repeat, and a 3-day sub-cycle [15]. SMAP SSS measurements have a similar spatial resolution (60 km) and an 8-day repeat with a 2–3-day sub-cycle [16]. Remote sensing is the only viable tool to monitor river plumes systematically. Compared with satellite observations, in-situ measurements are sparse in time and space, making it challenging to study the seasonal to interannual variability of river plumes, regions with sharp, transient horizontal gradients [13]. Global ocean models and assimilation products have a major limitation in representing SSS variability near river plumes because of the common use of river discharge climatology and relaxation of model SSS towards seasonal climatology [10].

Previous studies have evaluated the consistency of satellite SSS with point-wise in-situ measurements in some major river mouths such as those in the Gulf of Mexico and the Bay of Bengal [13,17,18]. However, the heterogeneous sampling of the in-situ measurements pose a difficulty in evaluating seasonal-to-interannual variations of SSS near river mouths. Gridded salinity products based on in-situ measurements [19] have the potential capability to depict these variations if there is reasonable sampling. However, the consistency of these in-situ gridded products, their consistency with satellite SSS, and the consistency of satellite SSS in representing SSS variations near major river mouths have not been examined systematically. To fill this knowledge gap, this study intends to systematically evaluate the consistency among two satellite and two in-situ gridded SSS products in characterizing seasonal-to-interannual variations of SSS within a few hundred kilometers of the mouths of the top 10 rivers in low- to mid-latitude oceans. We do not study SSS in high-latitude river mouths because of the lack of sensitivity of the current generation of satellite sensors to SSS in cold-water (<5° C) environment and the paucity of in-situ salinity measurements in these regions [20]. The results of this study have significant implications for scientific investigation of river plume variability using satellite and in-situ products, observing system design and enhancement, and ocean modeling an assimilation. We first introduce our datasets and method (Section 2). We then compare monthly, seasonal, and interannual variability of the different in-situ and satellite SSS products in the major river plume areas (Section 3). In Section 4, we further discuss the implications of our results to the sampling of in-situ and satellite observing systems and evaluating satellite products using in-situ products. Section 5 summarizes the findings.

## 2. Materials and Methods

Our analysis is carried out using various satellite and in-situ SSS measurements. Brief descriptions of the characteristics of various datasets are provided below. The sources of all datasets are listed in the acknowledgments.

### 2.1. Data

We use the Level-3 debiased version-5 SMOS SSS product produced by LOCEAN/IPSL (UMR CNRS/UPMC/IRD/MNHN) laboratory and ACRI-st company that participate in the Ocean Salinity Expertise Center (CECOS) of Centre Aval de Traitement des Donnees SMOS (CATDS) [15]. The 9-day running mean maps on an Equal-Area Scalable Earth (EASE) 25-km grid were provided from January 2010 to November 2020 every four days. These observations are averaged onto monthly intervals.

We also use the Level-3 version-4 SMAP SSS product distributed by Remote Sensing Systems (RSS), with a 70 km spatial resolution and an 8-day running mean. These measurements are available from May 2015 to present on a $0.25° \times 0.25°$ daily grid [16]. These observations are also averaged onto monthly intervals.

A total of two different in-situ-based gridded salinity products are used for comparison with satellite salinity data. The first one, from the Scripps Institution of Oceanography (SIO) is a monthly optimal interpolation product based only on Argo data, produced on an $1 \times 1°$ grid since 2004 [19]. The second one from Japan Agency for Marine-Earth Science and Technology (JAMSTEC) is based on Argo data, as well as buoy and Conductivity-

Temperature-Depth (CTD) measurements. This monthly product is delivered on a $1 \times 1°$ grid using an objective analysis since 2001 [21].

Data from the 2018 World Ocean Database (WOD 2018) were retrieved directly from NOAA's National Oceanographic Data Center (NODC) (now part of the National Centers for Environmental Information (NCEI) https://www.ncei.noaa.gov/access/world-ocean-database-select/dbsearch.html). Data from the WOD include conductivity/temperature/depth (CTD), gliders, profilers, and drifting buoys, and provide a unique opportunity for validation in a coastal or regional basin. Here, we use the location of salinity in-situ measurements from the 1–5 m depth range in order to have more in-situ near-surface observations. The WOD 2018 contains the full set of quality controls.

## 2.2. Method

We analyze the monthly, seasonal, and interannual variability of different SSS fields at the mouth of the 10 largest low to mid-latitude rivers which are the Amazon, Congo, Orinoco, Yangtze, Ganges/Brahmaputra, Mississippi, Parana/Plata, Mekong, Irrawaddy, and Columbia rivers according to [22]. We compute time series for SSS averaged over a $3° \times 3°$ box near each river mouth chosen based on the following criterion: the $3° \times 3°$ box closest to the coordinate of the river mouth location where all SSS products have at least half of the pixels with data. The $3° \times 3°$ is the nominal spacing of the Argo float array over the global ocean. However, near coastal regions the sampling could be sparser, thereby affecting the gridded products. For example, there is no SIO data at the mouth of the Yangtze and the Irrawaddy rivers and there is no JAMSTEC data at the mouth of the Mekong river. Figure 1 shows the location of each river mouth box. The coordinates of each box are presented in Table 1.

**Table 1.** Coordinates of each river mouth box shown in Figure 1.

|  | Latitude Boundaries | Longitude Boundaries |
|---|---|---|
| **Amazon** | [5.5°N, 8.5°N] | [51.5°W, 48.5°W] |
| **Congo** | [6°S, 9°S] | [10°E, 13°E] |
| **Orinoco** | [11.5°N, 14.5°N] | [62.5°W, 59.5°W] |
| **Yangtze** | [29.5°N, 32.5°N] | [124.5°E, 127.5°E] |
| **Ganges/Brahmaputra** | [17.5°N, 20.5°N] | [88.5°E, 91.5°E] |
| **Mississippi** | [26.5°N, 29.5°N] | [88.5°W, 85.5°W] |
| **Parana/Plata** | [38°S, 35°S] | [56°W, 53°W] |
| **Mekong** | [12.5°N, 15.5°N] | [109.5°W, 112.5°W] |
| **Irrawaddy** | [12.5°N, 15.5°N] | [94.5°E, 97.5°E] |
| **Columbia** | [44.5°N, 47.5°N] | [127.5°W, 124.5°W] |

From the Argo gridded and satellite SSS monthly fields, we compute seasonal and interannual variability time series at the mouth of each of the 10 rivers considered. Seasonal SSS maps are computed by averaging maps over the common period May 2015 to November 2020 each month of different years. The seasonal time series in each box is the average of these seasonal SSS maps within the box. The interannual variability time series is computed as a low-pass filter ($+/-$ 7-month running mean filter) on the non-seasonal SSS time series, the latter defined as the anomalies of the monthly fields from the seasonal climatology.

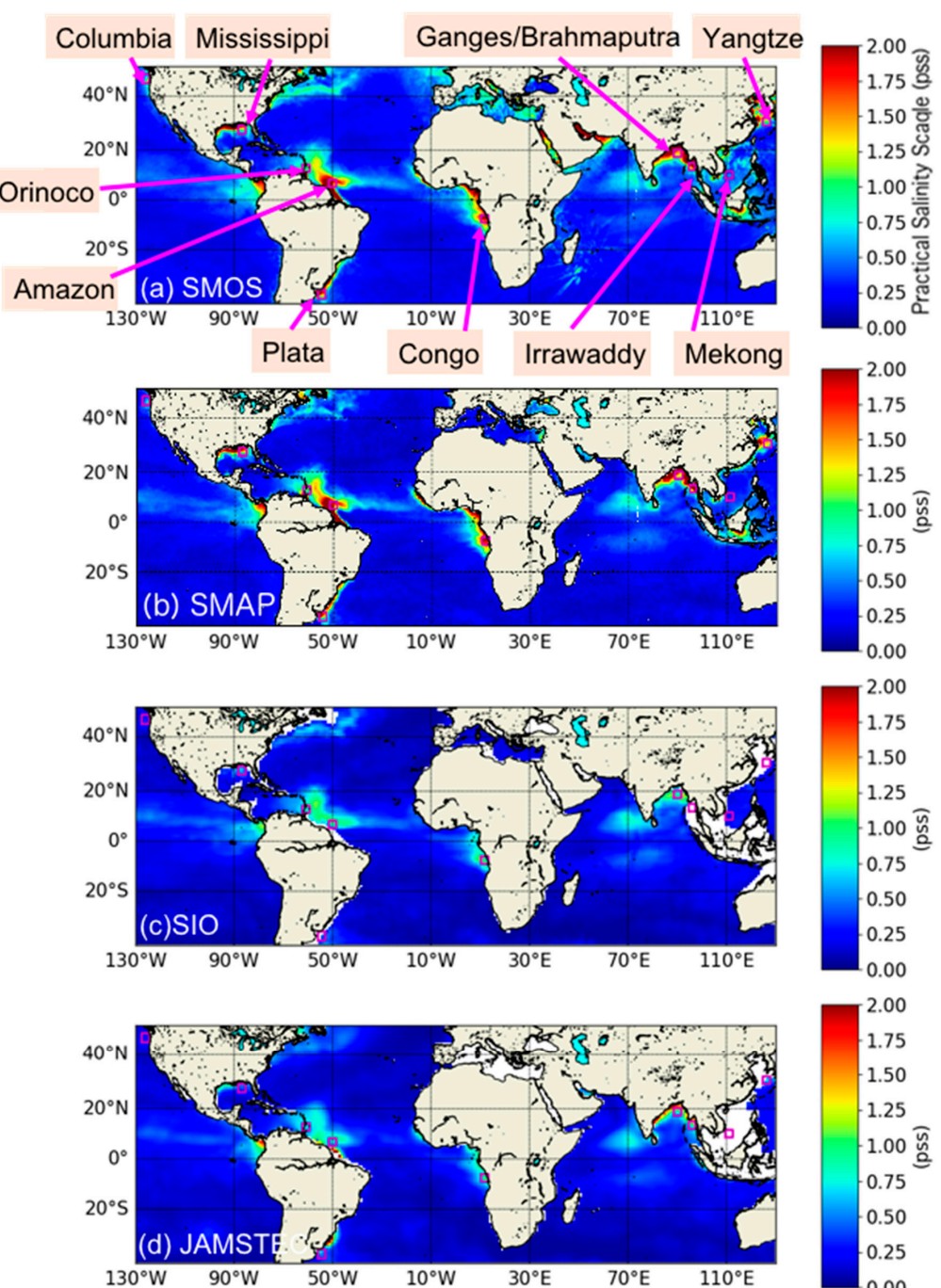

**Figure 1.** Standard deviation maps of (**a**) SMOS, (**b**) SMAP, (**c**) SIO, and (**d**) JAMSTEC SSS over January 2010 to November 2020 for (**a**), (**c**), and (**d**) and May 2015 to November 2020 for (**b**). The magenta squares represent the boxes at the 10 river mouths considered in this study (from left to right): Columbia, Mississippi, Parana/Plata, Orinoco, Amazon, Congo, Brahmaputra/Ganges, Irrawaddy, Mekong, and Yangtze.

## 3. Results

Figure 1 shows the SSS variability captured by each product globally over January 2010 to November 2020 for SMOS, SIO, and JAMSTEC and May 2015 to November 2020 for SMAP. All the maps show high variability of SSS in regions associated with major river plumes (e.g., Amazon, Congo, Ganges/Brahmaputra, Mississippi, Plata) and also along the western coast of Panama, south western coast of India in the Arabian Sea and the Gulf Stream. However, the amplitude of the variability is much higher in the satellite

products, especially in river plume regions. Indeed, the amplitude of the variability reaches for example more than 2 psu in the Amazon plume against about 1 psu in the in-situ-based gridded products. The standard deviation maps of SMOS and SMAP have similar magnitudes of variability. The in-situ-based products do not capture the Colombia River plume. Additionally, the SIO product do not capture the Plata River plume. We also notice the absence of data in the SIO and JAMSTEC products in the south east Asia seas.

Figure 2 shows August 2015 maps of SSS in the north west tropical Atlantic Ocean near the Amazon River mouth. All four products consistently capture the extent of the Amazon River plume. However, SMOS and SMAP both capture salinity gradients and features in the plume whereas the in-situ-based products do not. In particular, SMOS and SMAP capture the fresher waters (below 31 psu) near the river mouth that are transported along the North Brazilian Current retroflection in summer. In addition, due to the higher spatial resolution of satellite SSS, the edges of the plume are better captured by SMOS and SMAP than by SIO and JAMSTEC. In addition, SMOS and SMAP are able to retrieve data closer to the coast than the in-situ-based gridded products.

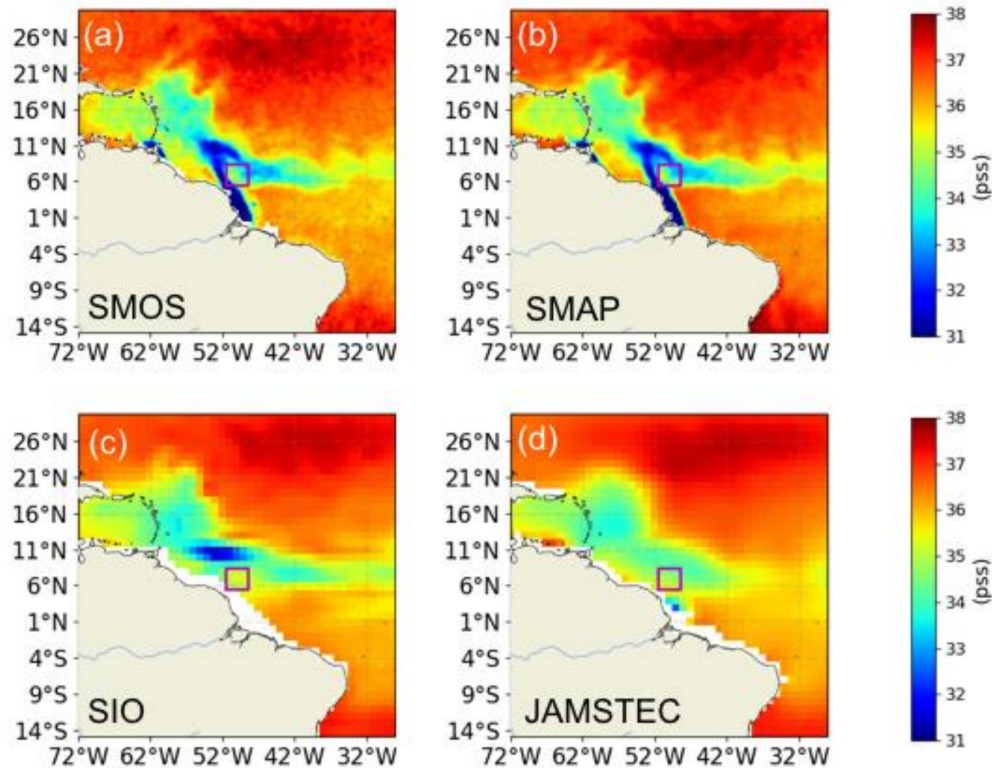

**Figure 2.** August 2015 monthly maps of (**a**) SMOS, (**b**) SMAP, (**c**) SIO, (**d**) JAMSTEC SSS near the Amazon mouth in the northwestern tropical Atlantic Ocean, 2–3 months after the peak discharge. The magenta square represents the box chosen to compute the SSS time series at the Amazon River mouth.

In Figure 3, we show the monthly time series at the mouth of the 10 rivers by averaging the SSS within the boxes defined in the method section and shown on Figure 1. The four different products all capture the seasonal freshening at each river mouth with similar timing except for the Plata River. In-situ-based products do not seem to capture any seasonality at the Plata River mouth because this river plume hugs near the coast, making it challenging for in-situ sensors to capture. For most river plumes where all the products capture the seasonality of the freshening, the amplitude of this freshening is not consistent between satellite and in-situ-based products. Table 2 summarizes the standard deviation values of SSS difference between two of the four products based on the monthly time series at each river mouth shown in Figure 3. For comparison purposes, the statistics are computed over the common period for the four products April 2015 to November 2020. It

shows that the standard deviation is typically much smaller between SMOS and SMAP (~0.1–0.5 psu) than that between in-situ-based and satellite products (~0.3–1.2 psu) and that between JAMSTEC and SIO products (~0.1–0.6 psu). The standard deviation between JAMSTEC and SIO is however typically smaller than that between the in-situ-based and satellite products.

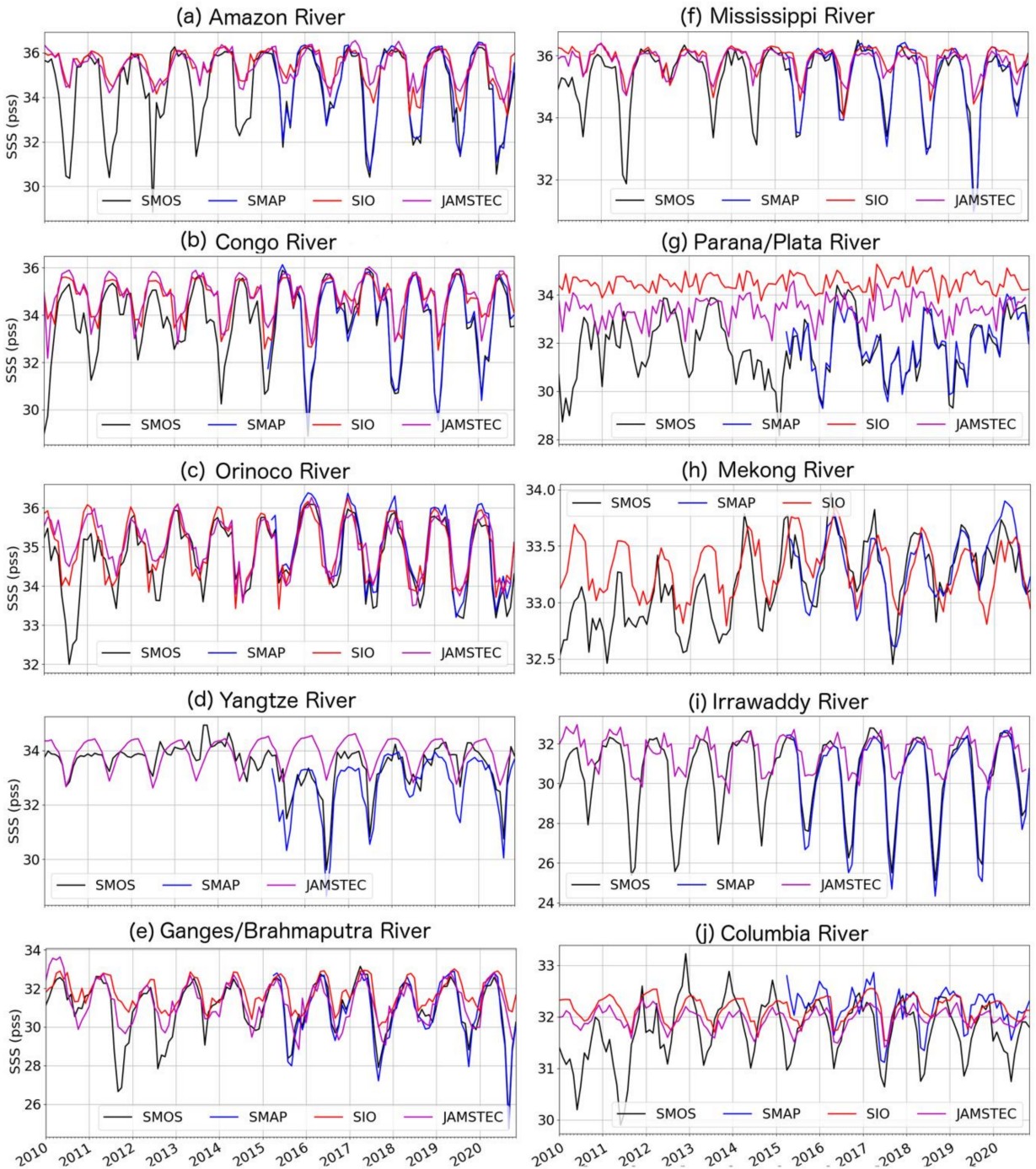

**Figure 3.** Monthly time series from January 2010 to November 2020 of SMOS (black), SMAP (blue), SIO (red), and JAMSTEC (magenta) SSS at the 10 river mouths represented by magenta squares in Figure 1.

**Table 2.** Standard deviation of monthly SSS difference between each two pairs of the four SSS products (SMOS, SMAP, SIO, and JAMSTEC) over their common period May 2015 to November 2020 at each river mouth.

| | STD (psu) | | | | | |
|---|---|---|---|---|---|---|
| | **SMOS/SMAP** | **SIO/JAMSTEC** | **SMOS/SIO** | **SMOS/JAMSTEC** | **SMAP/SIO** | **SMAP/JAMSTEC** |
| **Amazon** | 0.32 | 0.45 | 1.08 | 1.22 | 1.05 | 1.19 |
| **Congo** | 0.27 | 0.37 | 1.13 | 1.17 | 1.06 | 1.09 |
| **Orinoco** | 0.12 | 0.32 | 0.41 | 0.43 | 0.38 | 0.44 |
| **Yangtze** | 0.54 | N/A | N/A | 0.77 | N/A | 0.80 |
| **Ganges/Brahmaputra** | 0.38 | 0.59 | 1.00 | 1.03 | 1.16 | 1.16 |
| **Mississippi** | 0.22 | 0.26 | 0.78 | 0.80 | 0.88 | 0.91 |
| **Parana/Plata** | 0.40 | 0.27 | 1.21 | 1.28 | 1.19 | 1.25 |
| **Mekong** | 0.17 | N/A | 0.26 | N/A | 0.18 | N/A |
| **Irrawaddy** | 0.36 | N/A | N/A | 1.62 | N/A | 1.86 |
| **Columbia** | 0.34 | 0.12 | 0.41 | 0.39 | 0.30 | 0.32 |

From the monthly time series presented in Figure 3, we compute seasonal time series as explained in the Methods section. Figure 4 provides a better visualization showing that the four different products capture very well the seasonal freshening with similar timing as shown first in Figure 3, except for the Plata River. However, the amplitude of the seasonal cycle is typically much smaller in in-situ-based than satellite products. Both satellite products are well correlated on seasonal time scale with an excellent correlation coefficient (R) (Table 3). Both in-situ-based gridded products are consistent in depicting the amplitude of the seasonal cycle for all the rivers with also excellent correlation coefficients (Table 3) but it is usually much smaller than the amplitude detected by satellite, ranging from 2 psu smaller for the Amazon, Mississippi, and Congo rivers to 4 psu smaller for the Irrawaddy River. Satellite and in-situ are still well correlated at seasonal time scale but less than both satellite products and both in-situ products are Table 3. As mentioned before, the close proximity of this river plume to the coast prevented the in-situ-based gridded products to capture the SSS variability associated with the river plume. The freshening near the Orinoco and Mekong River mouths seems to be well captured by all the products on seasonal time scale. Near the Yangtze River mouth, SMOS and SMAP do not capture the same amplitude of the freshening, with that of SMOS being 1 psu fresher than that of SMAP. This might be due to the presence of Radio Frequency Interferences (RFI) in this region that had to be filtered in each product possibly leading to some errors. However, when the seasonal cycles are computed for the same period (April 2015 to November 2020; not shown), the root mean square difference (RMSD) between SMOS and SMAP in depicting the seasonal cycle improves from 0.9 to 0.57 psu. For the Ganges/Brahmaputra River plume, SMOS and SMAP SSS are very close to each other in terms of the seasonal cycle both in phase and magnitude. The magnitude of seasonal freshening is substantially weaker in SIO and JAMSTEC than in the satellite SSS. Moreover, the timing of the highest SSS and that of the lowest SSS in SIO and JAMSTEC products differ by 1 to 2 months. While the month of the highest SSS in JASMTEC agrees better than the satellite SSS, the month of the lowest SSS in SIO agrees better with the satellite SSS. The timing of the highest and lowest SSS between satellite and in-situ products differ by 1 to 2 months at the mouth of the Amazon, Yangtze, Mekong, Irrawaddy, and the Columbia River. The Columbia River has a much smaller plume than other rivers considered in the study (see Figure 1), the differences in the seasonality captured by all products are large. However, despite the time-mean offset between SMOS and SMAP, the difference in the magnitude of the seasonal variation is similar between SMOS and SMAP, and again, smaller in the in-situ products.

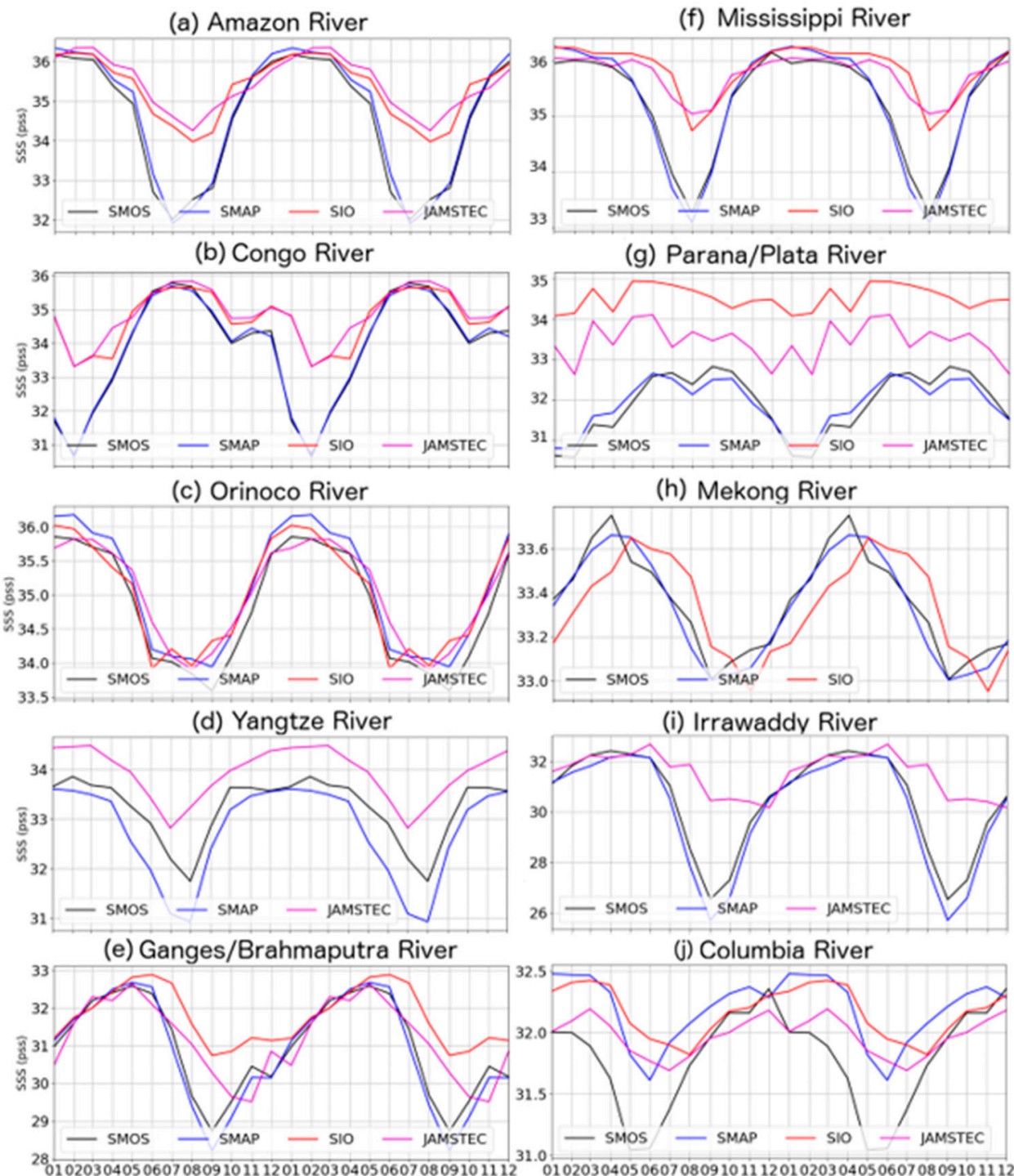

**Figure 4.** Seasonal variability time series from January 2010 to November 2020 (April 2015 to November 2020 for SMAP) of SMOS (black), SMAP (blue), SIO (red), and JAMSTEC (magenta) SSS at the 10 river mouths represented by magenta squares in Figure 1. For visualization purposes, the seasonal cycles are repeated twice.

**Table 3.** Correlation coefficient of seasonal SSS between each two pairs of the four SSS products (SMOS, SMAP, SIO, and JAMSTEC) over their common period May 2015 to November 2020 at each river mouth.

| | Correlation Coefficient | | | | | |
| --- | --- | --- | --- | --- | --- | --- |
| | **SMOS/SMAP** | **SIO/JAMSTEC** | **SMOS/SIO** | **SMOS/JAMSTEC** | **SMAP/SIO** | **SMAP/JAMSTEC** |
| **Amazon** | 1 | 0.95 | 0.97 | 0.91 | 0.97 | 0.93 |
| **Congo** | 1 | 0.95 | 0.86 | 0.90 | 0.86 | 0.90 |
| **Orinoco** | 1 | 0.94 | 0.96 | 0.97 | 0.98 | 0.97 |
| **Yangtze** | 0.97 | N/A | N/A | 0.91 | N/A | 0.96 |
| **Ganges/Brahmaputra** | 0.99 | 0.86 | 0.87 | 0.83 | 0.84 | 0.82 |
| **Mississippi** | 1 | 0.93 | 0.88 | 0.95 | 0.86 | 0.94 |
| **Parana/Plata** | 0.97 | 0.62 | 0.58 | 0.45 | 0.65 | 0.58 |
| **Mekong** | 0.97 | N/A | 0.71 | N/A | 0.76 | N/A |
| **Irrawaddy** | 1 | N/A | N/A | 0.72 | N/A | 0.70 |
| **Columbia** | 0.84 | 0.88 | 0.51 | 0.75 | 0.78 | 0.83 |

The interannual variabilities of SSS for the 10 river mouths are shown in Figure 5. The interannual variabilities of JAMSTEC and SIO SSS, when both available for the same river mouth, are usually well correlated, except for the Amazon River (Table 4). The interannual variabilities captured by the two satellites are also well correlated (Table 4). The correlation coefficients between in-situ and satellite products are however much lower (Table 4). JAMSTEC and SIO do not capture the interannual variability captured by both SMOS and SMAP.

**Table 4.** Correlation coefficient of interannual SSS between each two pairs of the four SSS products (SMOS, SMAP, SIO, and JAMSTEC) over their common period May 2015 to November 2020 at each river mouth.

| | Correlation Coefficient | | | | | |
| --- | --- | --- | --- | --- | --- | --- |
| | **SMOS/SMAP** | **SIO/JAMSTEC** | **SMOS/SIO** | **SMOS/JAMSTEC** | **SMAP/SIO** | **SMAP/JAMSTEC** |
| **Amazon** | 0.82 | 0.25 | 0.67 | 0.43 | 0.83 | 0.35 |
| **Congo** | 0.95 | 0.82 | 0.70 | 0.82 | 0.70 | 0.86 |
| **Orinoco** | 0.91 | 0.95 | 0.80 | 0.71 | 0.64 | 0.55 |
| **Yangtze** | 0.82 | N/A | N/A | −0.75 | N/A | −0.85 |
| **Ganges/Brahmaputra** | 0.81 | 0.83 | −0.02 | −0.03 | 0.41 | 0.44 |
| **Mississippi** | 0.97 | 0.71 | 0.57 | 0.65 | 0.65 | 0.63 |
| **Parana/Plata** | 0.73 | 0.62 | −0.64 | −0.18 | −0.70 | −0.35 |
| **Mekong** | 0.77 | N/A | 0.09 | N/A | −0.01 | N/A |
| **Irrawaddy** | 0.98 | N/A | N/A | 0.30 | N/A | 0.39 |
| **Columbia** | 0.54 | 0.93 | 0.09 | −0.16 | 0.39 | 0.25 |

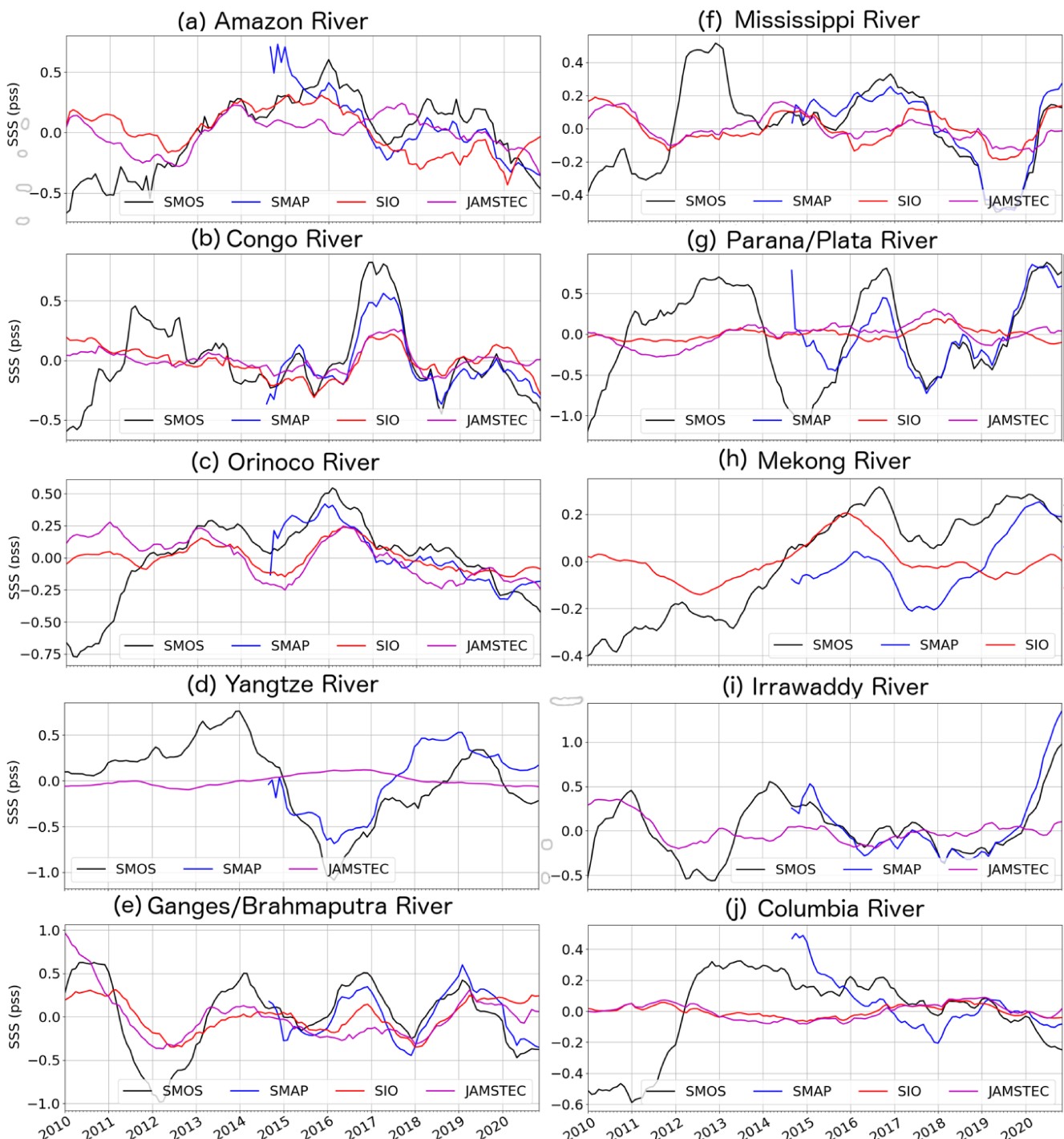

**Figure 5.** Interannual variability time series from January 2010 to September 2019 of SMOS (black), SMAP (blue), SIO (red), and JAMSTEC (magenta) SSS at the 10 river mouths represented by magenta squares in Figure 1.

## 4. Discussion

The findings from this study illustrate the excellent consistency between two independent SSS satellite products from SMOS and SMAP near the major low-latitude river mouths of the world ocean both for seasonal and interannual time scales. However, as shown in this study, satellite and in-situ gridded products tend to be more consistent during the high SSS than low SSS season. High SSS season follows the low-discharge season while low SSS season follows the high discharge season. The fact that the in-situ gridded products show larger discrepancy from the satellite SSS during low SSS season is believed to be caused by

the inadequate sampling of the in-situ measurements to capture the river plumes, which are associated with stronger spatial and temporal variability associated with the more active river plumes following the high-discharge season.

This study also shows that the difference between the two in-situ gridded products is often larger than the difference between the two satellite products. The SIO and JAMSTEC products are based on mostly the same in-situ measurements, with SIO only based on Argo measurements and JAMSTEC on Argo data, as well as buoy and CTD measurements. The two satellite products are from two independent satellites. This further underscores the issue with in-situ sampling. When in-situ sampling is inadequate, the effect of difference in gridding algorithms for the in-situ products become more accentuated in the gridded SSS product.

Figure 6 shows the 2019 in situ density maps in the Gulf of Mexico near the Mississippi River mouth and in the Bay of Bengal near the Ganges/Brahmaputra River during the low discharge and high SSS season and the high discharge and low SSS season with the river plume contour each month, respectively. This figure shows that the SSS has more variability during the low SSS season than during the high SSS season. Additionally, the figure shows that the Argo distribution is inadequate to sample the plume.

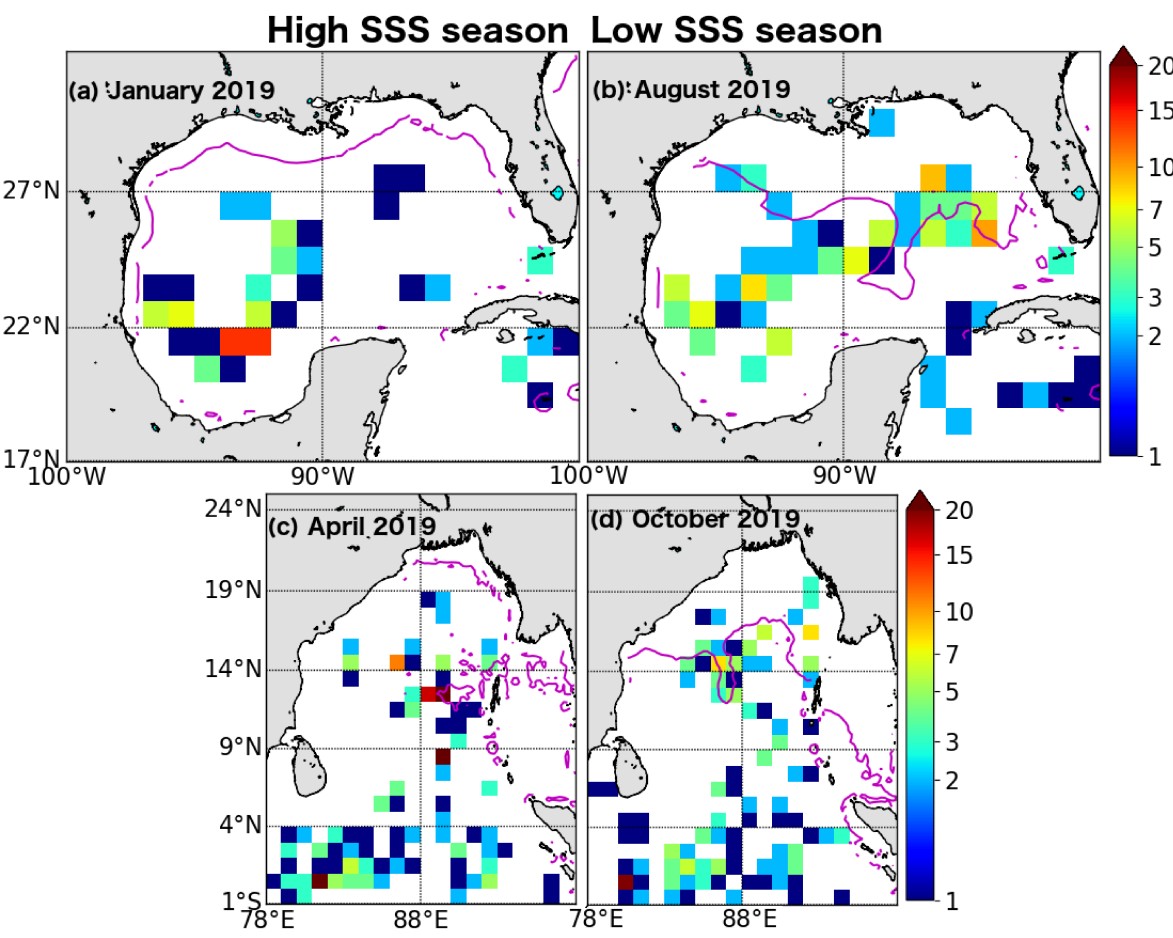

**Figure 6.** Number of 0–5 m deep in-situ salinity measurements within $1 \times 1$ degree bins over a month in the Gulf of Mexico (**a,b**) and Bay of Bengal (**c,d**) during the high SSS season (January and April 2019, respectively) and the low SSS season (August and October 2019, respectively). The plume contours (35 psu in the Gulf of Mexico and 32 psu in the Bay of Bengal, computed using monthly SMAP SSS data) are represented in magenta.

Reference [23] computed a global evaluation of NASA/SAC-D Aquarius SSS using two in-situ products, one of them also used here (SIO). Large differences are found near

several major river plumes between the SIO and University of Hawaii (UH; another monthly $1 \times 1°$ gridded Argo product) products in the Amazon, Ganges/Brahmaputra, and Congo river plume regions that can be larger than the difference between Aquarius and in-situ. Therefore, in evaluating gridded satellite SSS using gridded in-situ products, it is important not to use the difference between satellite and in-situ gridded products as the uncertainty of the former in these regions. In addition, in evaluating globally or regionally averaged uncertainties of satellite SSS using gridded in-situ SSS products, river plume regions should be excluded in the global statistics.

Furthermore, [24] showed that the magnitude and phase of seasonal SSS from in-situ gridded products (SIO and EN4, the version 4 of the Met Office Hadley Centre "EN" series of gridded data sets of global quality-controlled ocean salinity profiles) in major river plumes regions are subject to significant errors, and should not be used as benchmark to evaluate the assess the accuracies of the satellite SSS in these regions.

## 5. Conclusions

In this study, we show that SSS from both SMOS and SMAP satellites have excellent consistency in depicting seasonal and interannual SSS variations near major river mouths. Even when both in-situ-based gridded products are consistent with each other, they tend to underestimate the seasonal and interannual variation of SSS substantially within a few degrees of the river mouth. Moreover, the difference between the two in-situ gridded products can be larger than the difference between the two satellite products. The differences between the gridded in-situ and satellite products tend to be larger during the low SSS season. This is attributable to the inadequate sampling of the in-situ measurements in capturing the variability of the river plumes that are stronger during the low SSS season.

The results of this study have significant implications for the scientific studies of river plumes, in-situ and satellite salinity observing systems, and modeling and assimilation. Coastal oceans near river plumes are associated with strong spatial and temporal variability of SSS, especially following the high-charge season when the plumes are the strongest and SSS lowest. This poses a significant challenge to the in-situ observing system even for a spatial scale that is the nominal spacing of the Argo array ($3° \times 3°$). The extremely sparse distributions of in-situ measurements near river mouths (e.g., Figure 6) preclude a proper quantification of the absolute uncertainties of satellite SSS in these regions. Moreover, the in-situ data distributions shown in Figure 6 suggest that there can be little to no in-situ measurements within a few hundred kilometers of the river mouths during certain months. Therefore, the spatiotemporal variability of SSS near river mouths as represented by the in-situ based gridded products are heavily influenced by spatiotemporal extrapolation and interpolation of the sparse point-wise measurements (often far away from river mouths). The representation of SSS variability near river mouths by the in-situ based gridded products should thus be treated with caution. On the other hand, the excellent consistency of SSS measurements from two independent satellites (SMOS and SMAP) improves the confidence in the abilities of the satellite SSS to depict river plume variability. In fact, high-resolution (~ a few km) ocean color measurements from independent satellites also provide consistent characterization of river plume structure on the coarser scales (~50 km) resolved by satellite SSS [9,25,26].

Previous studies (e.g., [10]) have showed that state-of-the-art global ocean models and assimilation products underestimated the seasonal and interannual variability of SSS near the Mississippi River plume in part because of the strong relaxation of model SSS to gridded salinity climatology that is inadequate to represent seasonal variability of river plumes, let alone the suppression of interannual SSS variability in the model due to the relaxation of model SSS to seasonal SSS climatology. The seasonal and interannual variability of SSS near major river mouths around the world ocean reported in our study, consistently depicted by SMAP and SMOS, provide useful observations to constrain the models and assimilations

**Author Contributions:** Conceptualization, T.L., S.F.; methodology, T.L., S.F; validation, S.F.; formal analysis, S.F.; investigation, S.F.; data curation, S.F; writing—original draft preparation, S.F.; writing—

review and editing, T.L.; funding acquisition, T.L. All authors have read and agreed to the published version of the manuscript.

**Funding:** The research described in this paper was carried out at the Jet Propulsion Laboratory, California Institute of Technology, under a contract with NASA.

**Acknowledgments:** The research described in this paper was carried out at the Jet Propulsion Laboratory, California Institute of Technology, under a contract with NASA. Data for this paper are available at the following data centers we gratefully thank: the Ocean Salinity Expertise Center (CECOS) of the CNES-IFREMER Centre Aval de Traitement des Donnees SMOS (CATDS), at IFREMER, Plouzane (France) for SMOS SSS data (https://www.catds.fr/Products/Available-products-from-CEC-OS/CEC-Locean-L3-Debiased-v5) accessed on 15 December 2020; the Physical Oceanography Distributed Active Archive Center (PO.DAAC) for SMAP SSS data (https://podaac.jpl.nasa.gov/dataset/SMAP_JPL_L3_SSS_CAP_8DAY-RUNNINGMEAN_V4) accessed on 15 December 2020; the Scripps Institution of Oceanography (SIO) for the Argo salinity gridded product (http://www.argo.ucsd.edu/Gridded_fields.html) accessed on 15 December 2020; the Japan Agency for Marine-Earth Science and Technology (JAMSTEC) for the Argo salinity gridded product (https://pubargo.jamstec.go.jp/public/MOAA_GPV/Glb_PRS/OI/) accessed on 15 December 2020; the Research Data Archive at the National Center for Atmospheric Research (NCAR) for the river discharge data (https://rda.ucar.edu/datasets/ds551.0/#!description) accessed on 15 December 2020.

**Conflicts of Interest:** The authors declare no conflict of interest. The funders had no role in the design of the study; in the collection, analyses, or interpretation of data; in the writing of the manuscript, or in the decision to publish the results.

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
