# Peer review of "Seasonal and Interannual Variability of Sea Surface Salinity Near Major River Mouths of the World Ocean Inferred from Gridded Satellite and In-Situ Salinity Products"

_remotesensing, doi:10.3390/rs13040728_

Round 1

Reviewer 1 Report

The authors used the two sets of satellite data of SMOS and SMAP to observe the variations in sea surface salinity outside the world's top 10 estuaries and compared them with the two sets of field observation data. The most important contribution of this manuscript is that the authors found that satellite salinity data can successfully observe fresh water during flood and dry periods, which is difficult to observe with field observation data. I think this manuscript is very valuable, especially for researchers who study the changes and mechanisms of ocean physical processes near the estuary. Below I find that there are some small suggestions that need to be clarified.

Minor comments:

  1. Please confirm whether the title of Reference 4 is correct.
  2. Please spell out the full name of "PSS (Pactical Salinity Scale)" on the colorbar in Figure 1.
  3. Figure 4, the authors point out that the time series is seasonal variability time series from January 2010 to November 2020 (April 2015-November 2020 for SMAP), but it seems that the time of the image should be only two years, which period?
  4. Page 5 & Figure 5, “The seasonal time series in each box is the average of these seasonal SSS maps within the box. The interannual variability time series is computed as a low-pass filter (+/- 7-month running mean filter) on the non-seasonal SSS time series, the latter defined as the anomalies of the monthly fields from the seasonal climatology.”, Does Figure 5 show the monthly anomalies or each monthly anomalies? In addition, why use the 7 month low-pass filter? What is the non-seasonal SSS series?
  5. What data is the plume contours drawn in Figure 6?
  6. What is the UH and EN4 products in section Discussion? Please spell out the full name.

Reviewer 2 Report

The paper is devoted to investigation of sea surface salinity (SSS) in the river mouth areas based on several sources: satellite observations and in-situ gridded measurements. The authors investigate some SSS characteristics in the areas of interest: mean state, variance, time series on montly, seasonal and interannual scales. It was shown that although if both in-situ-based gridded products are consistent with each other, they underestimate the seasonal and interannual variation of SSS substantially in some vicinity to the river mouths. Also, the difference between the two in-situ gridded products can be larger than the difference between the two satellite products. The differences between the in-situ and satellite products increases during the low SSS season. This may be due to the inadequate sampling of the in-situ measurements in cap-turing the variability of the river plumes that are stronger during the low SSS season.

The paper is of a high scientific and practical significance. It is always importan to compare satellite and in-situ data, estomate their distinct, quality, completeness e t. c., epecially in the river mouths with high variabitity, where the precision of data can be low. The level of English language is high enough, as well as readability of the text.

In general, I think that the paper can be published in the present form. I have just one moment for the authors to explain. As far as I could understand, they declare the satellite observations as true data, and comparing the in-situ measurements to them, they conclude that the latter are worse (underestimate SSS variation). Certainly, it is wrong. Every method (satellite observation or in-situ measurement) has its own disadvantages. In this case, it would be interesting to point at the advantages of the in-situ data comparative to satellite-based ones. I hope that the authors will explain, what they mean: in the response to reviewer, or, if needed, in the body of the paper.
